# From Similarity to Vulnerability: Key Collision Attack on LLM Semantic Caching

**Zhixiang Zhang**[1]  **Zesen Liu**[1]  **Yuchong Xie**[1]  **Quanfeng Huang**[2]  **Dongdong She**[1]

## Abstract

Semantic caching has emerged as a pivotal technique for scaling LLM applications, widely adopted by providers including AWS and Microsoft. By utilizing embedding vectors as cache keys, this mechanism effectively minimizes latency and redundant computation for semantically similar queries. In this work, we conceptualize semantic cache keys as a form of fuzzy hashes. We demonstrate that the locality required to maximize cache hit rates fundamentally conflicts with the cryptographic avalanche effect necessary for collision resistance. Our conceptual analysis formalizes this inherent trade-off between performance (locality) and security (collision resilience), revealing that semantic caching is inherently vulnerable to key collision attacks. While prior research has focused on side-channel and privacy risks, we present the first systematic study of integrity risks arising from cache collisions. We introduce *CacheAttack*, an automated framework for launching black-box collision attacks. We evaluate *CacheAttack* in security-critical tasks and agentic workflows. It achieves a hit rate of 86% in LLM response hijacking and can induce malicious behaviors in LLM agent, while preserving strong transferability across different embedding models. A case study on a financial agent further illustrates the real-world impact. Finally, we discuss mitigation strategies, highlighting a persistent trade-off between cache efficiency and robustness.

## 1. Introduction

Real-world LLM agents often need to process repeated LLM queries for different users. (Zaharia et al., 2024). These

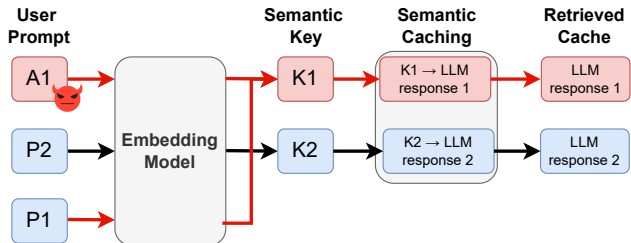

*Figure 1.* A brief overview of key collision in semantic caching. Semantically different $A_1$ (attacker) and $P_1$ (victim) unexpectedly maps to the same semantic key $K_1$. As a result, victim response is hijacked by the attacker. We show collision in **red lines**.

repeated queries incur significant redundant computation and inference latency. To mitigate this redundancy, modern LLM agents increasingly rely on caching (Gim et al., 2023; Zheng et al., 2024a). Beyond the transfer-internal KV cache (Zhao et al., 2023), the application-layer cache can store previously computed intermediate results and reuses them when similar queries are submitted by other users.

Semantic caching is a widely adopted mechanism in the application layer (e.g., AWS (AWS, 2025) and Microsoft (Microsoft, 2025a;b)): it embeds each user query into a semantic vector and reuses it when satisfying the matching condition. This fuzzy-match approach improves hit rates across tenants compared with exact-match approach and reduces both computational cost and inference latency (Bang, 2023; Microsoft, 2025a). Semantic caching commonly appears in two forms: (i) semantic cache(Bang, 2023; Regmi & Pun, 2024; Yan et al., 2025; Schroeder et al., 2025; Gill et al., 2025a;b; Li et al., 2024; Couturier et al., 2025), which caches and serves *final responses* via embedding similarity, and (ii) semantic KV cache(Zhu et al., 2025; Zhao & Mastorakis, 2025), which caches and reuses *KV states* indexed by semantic keys, enabling approximate reuse at the level of model execution.

While caching improves efficiency, it also exposes a unique attack surface of LLM agents. Previous security about LLM caching has been studied through privacy leakage, including prompt leakage and reconstruction (Wu et al., 2025; Luo et al., 2025; Gu et al., 2025; Song et al., 2025; Zheng et al., 2024b) via timing-based side channel in multi-tenant setting.

[1]Department of Computer Science and Engineering, The Hong Kong University of Science and Technology [2]Fudan University. Correspondence to: Dongdong She <dongdong@cse.ust.hk>.

*Proceedings of the 43rd International Conference on Machine Learning*, Seoul, South Korea. PMLR 306, 2026. Copyright 2026 by the author(s).

However, the integrity compromise in semantic caching remains largely underexplored.

We observe that semantic caching introduces an intrinsic vulnerability by design. The semantic key matching can be modeled as a locality-preserving fuzzy hash: it produces the same hash for similar queries. It fails to satisfy the *avalanche effect*, a security property of hash to ensure collision resistance. As a result, semantic caching is naturally vulnerable to cache collision. An attacker can easily leverage an adversarial example in the embedding model to trigger false-positive cache hits (i.e., malicious cache collisions). As illustrated in Figure 1, an adversary can intentionally craft adversarial prompts $A_1$ that have different semantics from the benign prompt $P_1$, yielding the same semantic key $K_1$ as $P_1$. An attacker can therefore *hijack* the LLM's response to the victim by retrieving an arbitrary cached response under his control.

This security risk is further amplified in LLM agent pipelines. The cached intermediate outputs often influence subsequent planning and tool invocation, so a single hijacked LLM response can redirect the agent's trajectory and induce cascading errors across downstream steps.

*CacheAttack*. Based on this vulnerability, we propose *CacheAttack*, the first key collision attack to semantic caching in a multi-tenant setting of LLM agents. The core of *CacheAttack* is a *generator-validator* framework designed to search for adversarial suffixes that induce false-positive cache key collisions efficiently. We introduce two distinct attack variants: (i) *CacheAttack*-1, which performs direct validation on the black-box target model but is time-consuming and prone to detection due to temporal dependency; (ii) *CacheAttack*-2, which which uses a surrogate model to emulate the target model to mitigate these issues.

**Evaluation.** In evaluation, we first evaluate *CacheAttack* in inducing LLM response hijacking. To achieve this, we construct an indirect prompt injection (IPI) dataset with 4,185 IPI prompts called SC-IPI, where *CacheAttack* attains high Hit Rate and Injection Success Rate. For an agentic scenario, the tool invocations are hijacked by reusing a malicious cached query, yielding high hit rate of 90.6% together with a substantial drop in answer accuracy, thereby demonstrating cascading errors. Besides, we also show that this attack transfers across different embedding models. We further quantify a trade-off between efficiency and robustness, through systematic sensitivity and generalizability analysis. To highlight practical severity, we further demonstrate end-to-end impact in a financial agent case study, where unintended malicious order execution is triggered, leading to potential economic loss. Finally, we investigate defenses and further show the persistent performance–security trade-off.

**Contribution.** We list our contributions as follows:

- We model semantic key matching in semantic caching as a *locality-preserving fuzzy hash function*.
- We show that key-collision attacks are an inherent integrity risk of semantic caching, rather than an anomaly. To study this risk systematically, we formalize a threat model and link this risk to the trade-off between locality and collision resilience.
- We propose *CacheAttack*, the first cache key collision attack framework against semantic caching, where we overcome the temporal dependency challenge inherent in black-box setting. Our code is available in `https://github.com/Zzx1011/CacheAttack`.
- We empirically demonstrate *CacheAttack* can induce both LLM response hijacking and agent tool invocation hijacking, thus leading to cascading errors.

## 2. Background and Related Works

### 2.1. Caching Mechanism in LLM

Caching has been widely used in LLM to reduce high computational cost and latency (Zhou et al., 2024). One common example is the key-value (KV) cache, which provides a performance boost at the decoding phase of LLM inference. It stores the KV states of attention layers for previously processed tokens, avoiding repeated computation of these cached KV states when generating the next token (Zhao et al., 2023).

LLM prompt cache and semantic cache leverage a *hash-table* design to reduce redundant computational cost when processing repeated or semantically similar user queries (Gim et al., 2023; Zheng et al., 2024a; Bang, 2023). They typically maintain a fast local storage of key-value pairs $(k, v)$. They compute a hash of the user query $k$ and store the corresponding LLM computation result $v$ in the cache storage. When the hash of a future user query matches the stored hash $k$, the corresponding $v$ is retrieved to avoid repeated computation. Prompt cache (Gim et al., 2023; Zheng et al., 2024a) uses the user prompt prefix as a cache key and stores the corresponding KV-state as a cache value. In addition to the exact user prompt prefix, semantic KV cache like SemShareKV (Zhao & Mastorakis, 2025) and SentenceKV (Zhu et al., 2025) show that a semantic embedding vector of user prompt can also serve as a cache key, improving cache hit rates and performance.

The semantic cache stores the semantic embedding vector of the user prompt as cache key and the corresponding LLM-generated response as cache value (Bang, 2023; Regmi & Pun, 2024; Yan et al., 2025; Schroeder et al., 2025; Gill et al., 2025a;b; Li et al., 2024; Couturier et al., 2025). It allows an approximate reuse of LLM-generated responses

for semantically similar user queries. Notably, semantic caches are often adopted and deployed by the real-world LLM service provider (such as AWS (AWS, 2025) and Microsoft (Microsoft, 2025a;b)) in the cross-tenant setting, so as to cut the computation cost for a huge volume of LLM user queries in practice (Bang, 2023; Regmi & Pun, 2024).

## 2.2. Hash Collision

A hash function maps data $x$ of arbitrary size into fixed length key $k$, denoted as $k = h(x)$. A hash collision occurs when two distinct data sets (say $x$ and $y$, where $x \neq y$) accidentally share the same key ($h(x) = h(y)$) (Preneel, 1993). Because the output space of a hash function is smaller than the input space, the hash collision is theoretically unavoidable. In a simple hash table, a hash collision can lead to worst-case runtime overhead during the hash table lookups (Maurer & Lewis, 1975). In practice, it can lead to catastrophic security failures such as password cracking, where an attacker can access a victim's account without a password (Oechslin, 2003), and critical certificate impersonation, where an attacker can impersonate a trusted victim to certify malware as benign. (Stevens et al., 2007).

## 2.3. Avalanche Effect

An ideal hash function should minimize hash collisions so as to mitigate performance degradation and further security risks. To achieve this collision-resilient feature, a practical hash function should satisfy a key security property: the *avalanche effect*. It defines that a small change $\epsilon$ in the input $x$ causes a significant and unpredictable change in the output, denoted as $\mathrm{Distance}(h(x), h(x + \epsilon)) \to \max$ (Upadhyay et al., 2022). For example, the Strict Avalanche Criterion requires that flipping just one input bit should cause every output bit to flip with a 50% probability (Webster & Tavares, 1985). The avalanche effect is a desired property for a hash function. Because it ensures that even minor variations in the input yield vastly different hash values, making it difficult for attackers to manipulate or predict the hash function's output. On the contrary, a hash function that shows poor avalanche effect is prone to hash collision (Upadhyay et al., 2022).

## 2.4. Prior Work and Our Position

Prior literature on the security risks of LLM caching systems primarily focuses on privacy leakage (Song et al., 2025; Zheng et al., 2024b; Gu et al., 2025), like the recovery of private user prompts (Wu et al., 2025; Luo et al., 2025) via timing-based side channel attack. While our work focuses on LLM response hijacking through cache key collision. In parallel, poisoning-style attacks (e.g., PoisonedRAG (Zou et al., 2025)) manipulate the external knowledge/content digested by LLM with corrupted retrieved documents or vector embedding. Unlike their manipulation of external content/knowledge, our work exploits the key collision in the LLM's semantic cache.

Modern semantic caching (including semantic cache and semantic KV cache) leverages the semantic embedding of the user query as the cache key. The cache key matching is then determined by a similarity score between two embedding vectors. Although this fuzzy semantic-level matching can increase cache hit rates compared to exact token-level matching, it exposes a new attack surface at the cache key matching layer. An adversary can craft a malicious user query whose cache key collides with a benign user query's cache key, thereby triggering unintended reuse of a malicious response under the adversary's control. Crucially, the attacker does not overwrite the cache value; they manipulate which cache value is reused by inducing key collisions in the LLM semantic cache.

We position our work as the *first systematic study* of key collision attacks against semantic caching. While some concurrent works (Wu et al., 2026a;b) observe empirical semantic cache poisoning or risks, we are the first to formalize this phenomenon as a key collision attack and unveil the inherent theoretical trade-off between locality and collision resistance. We show that the key collision attack can lead to LLM response hijacking and further propagate to LLM-powered agents as cascading errors.

# 3. Semantic Keys as Fuzzy Hashes: Locality vs. Collision Resistance

We show that the semantic keys of LLM semantic caching (including semantic cache and semantic KV cache) are essentially *fuzzy hashes* (Lee & Atkison, 2017). We then provide a conceptual analysis of the trade-off between locality (performance) and collision resistance (security) behind the semantic key's design. Our analysis reveals that the LLM semantic caching is inherently vulnerable to hash (i.e., cache key) collision attacks.

Formally, a user prompt $p$ is embedded into a semantic key as $k_p = f(p)$, where the embedding model is defined as $f : p \to \mathbb{R}^d$ and $\mathbb{R}^d$ denotes the continuous vector space. In semantic cache, the matching condition is determined by a similarity score between user semantic keys, as follows:

$$\mathrm{match}(p_1, p_2) = \begin{cases} \text{true,} & \text{if } \mathrm{sim}(k_{p_1}, k_{p_2}) \geq \tau \\ \text{false,} & \text{if } \mathrm{sim}(k_{p_1}, k_{p_2}) < \tau \end{cases} \quad (1)$$

. $\mathrm{sim}(\cdot, \cdot)$ denotes the similarity score of two semantic keys and $\tau$ is the similarity threshold. $p_1$ matches $p_2$ when the similarity score is equal to or greater than $\tau$ and vice versa. In semantic KV cache, the matching condition is often computed by a discrete semantic indexing rule such as *Locality-Sensitive Hashing (LSH)(Jafari et al., 2021)*, as

follows:

$$\text{match}(p_1, p_2) = \begin{cases} \text{true,} & \text{if } \ell(k_{p_1}) = \ell(k_{p_2}) \\ \text{false,} & \text{otherwise} \end{cases} \quad (2)$$

, where $\ell(\cdot)$ represents a classification function that partitions the continuous embedding space into several hyperplanes (Charikar, 2002).

Both cases can be considered fuzzy hashing, where multiple semantically similar user queries are mapped to the same bucket, i.e., the same semantic cache key. This strong locality feature sharply differs from the security property of conventional hash functions: the avalanche effect, as we explain in Sec 2.3. Specifically, given two semantically similar user prompts $p_1$ and $p_2$, under the fuzzy semantic key setting, they share the same hash value, as $\text{Distance}(h(p_1), h(p_2)) \to 0$. While the avalanche effect requires the difference in hash values for two similar user queries to be sufficiently large, as $\text{Distance}(h(p_1), h(p_2)) \to \max$, to ensure strong collision resistance.

This conflict reveals a hidden dilemma behind the semantic key's objective: a trade-off between locality (performance) and collision resistance (security). On the one hand, semantic keys are designed to preserve *locality* so as to maximize cache hits for semantically similar user queries. On the other hand, this design choice necessarily creates a *coarse-grained match criterion* (either through a permissive threshold $\tau$ or coarse partitioning), which is exactly the opposite of an avalanche-effect separation that ensures strong collision resistance.

To formalize this trade-off, we provide a theoretical bound on the false-positive risk inherent in this design. Let $\mathcal{B}$ denote the set of queries for which returning a cached response $p_c$ is semantically incorrect for a given query $p$. The probability of a false-positive hit is lower bounded as follows:

**Lemma 3.1** (Lower Bound on False-Positive Hits). *Given a cache hit probability $h = \Pr[\text{match}(p, p_c) = \text{true}]$, the probability of a false-positive hit is lower bounded by:*

$$\Pr[p \in \mathcal{B} \mid \text{match}(p, p_c) = \text{true}] \geq 1 - \frac{\Pr[p \notin \mathcal{B}]}{h} \quad (3)$$

Detailed proof is provided in Appendix A.

This trade-off further introduces an intrinsic vulnerability in the LLM semantic caching: hash collisions(i.e., semantic key collision). An attacker can easily craft an adversarial example in the semantic embedding space (Alzantot et al., 2018; Ebrahimi et al., 2018) for a false-positive hash collision. Unlike true-positive hash collisions, where queries share similar semantics, false-positive hash collisions can be arbitrary matches between any queries. Hence, the attacker can hijack a user query with arbitrary malicious instructions via such an unexpected and malicious cache collision.

## 4. Threat Model

In this section, we model the key collision attack as an adversarial example in the embedding model of the semantic cache. Further, the attacker leverages the cache key collision to hijack LLM response and induce malicious behaviors in the LLM agent.

### 4.1. Attack Target

Semantic caching is deployed as a critical middleware within LLM agent frameworks in multi-tenant settings. When a user query arrives, the system uses an embedding model to convert the prompt into a vector, which serves as the semantic cache key. It then determines whether any existing cache entry can be reused based on a predefined decision boundary. If a cache hit occurs, the agent directly retrieves and executes the stored execution *plans* and *tool-invocations*, skipping the costly LLM inference. In this entire system, the embedding model of semantic caching is our attack target.

### 4.2. Attacker's Knowledge and Capabilities

We assume the attacker can send arbitrary user prompts to the LLM system and receive the corresponding response. We specifically assume the embedding model that computes the cache key is a black-box function $f(p)$ to the attacker, which means its parameters, specific vector representations, and similarity threshold score are all inaccessible to the attacker. Under these black-box constraints, the attacker can optimize adversarial prompts against a local surrogate embedding model that mimics the key generation of the target model and infer cache hit/miss by analyzing LLM's observable signals. These signals include response consistency, recurring generation patterns, and variations in the end-to-end LLM response latency. Attacker can easily use the publicly released embedding models (e.g., BAAI/bge-small-en-v1.5(BAAI, 2024)) as surrogate models to launch the attack.

We emphasize that the attacker can select *arbitrary* prompt as their collision targets. Unlike prior poisoning works that assume the attacker can directly modify cached values (Zou et al., 2025), our attack neither modifies existing cached values nor accesses internal KV representations. Additionally, we don't control cache eviction policies or alter the embedding model's or the underlying LLM's parameters. All attacks are conducted at inference time and exploit the locality property of semantic caching rather than the backend LLM's internal decision boundaries.

### 4.3. Attacker's Goals

The attacker's goal is to induce *response hijacking* by triggering false-positive cache hits via semantic key col-

lisions. Given a benign user prompt $p_{\text{user}}$, the attacker seeks to construct an adversarial prompt $p_{\text{adv}}$ that satisfies the cache matching condition for $p_{\text{user}}$, denoted as $\text{match}(p_{\text{user}}, p_{\text{adv}}) = \text{true}$ detailed in Eq. (1) and Eq. (2). While $p_{\text{adv}}$ includes malicious instructions controlled by the attacker and is semantically dissimilar to $p_{\text{user}}$. This results in a semantic misalignment between the user's intent and LLM response, a phenomenon we refer to as *response hijacking*.

Our LLM response hijacking arises from false-positive cache hit, rather than common prompt injection (Liu et al., 2024) or backend LLM hallucination (Ji et al., 2023). The retrieved response can semantically deviate from the victim's original intention to malicious content under the attacker's control and further manifest as *misinformation*, *misleading advice*, or even explicitly *malicious content*, depending on the content of the cached LLM response, as we will show in Section 6. The attacker's ultimate goal is to leverage such hijacked responses to induce harmful behavior in real-world LLM agents.

## 5. Methodology

In this section, we present an automated framework called *CacheAttack*. We first formalize our adversarial example generation as a generator–validator framework, and then propose two concrete instantiations: *CacheAttack*-1 and *CacheAttack*-2.

### 5.1. Generator-Validator Framework

*CacheAttack* is to generate an adversarial prompt $p_a$ that collides with an arbitrary victim prompt $p_v$ under the matching condition of LLM semantic caching.

**Generator.** We design an adversarial prompt suffix generator that can lead to a collision with the victim prompt. Formally, we construct attacker queries in the form $p_a = p_{\text{src}} \oplus s$, where $p_{\text{src}}$ is the source prompt and $s \in \mathcal{V}^L$ is a discrete suffix. We optimize $s$ via a GCG-based (Zou et al., 2023) search by trading off collision strength and adversarial payload fluency:

$$\min_{s \in \mathcal{V}^L} \mathcal{L}_{\text{col}}(p_{\text{src}} \oplus s, \ p_v) \ + \ \lambda \, \text{PPL}_{\mathcal{M}}(p_{\text{src}} \oplus s), \quad (4)$$

where $\mathcal{M}$ is a language model used only for Perplexity (PPL) scoring and $\lambda > 0$ controls the trade-off. Here, we include perplexity to ensure a stealthy, robust attack that can bypass the LLM agent's input filters, with a detailed discussion in Section 7.2.

For semantic cache, the collision loss $\mathcal{L}_{\text{col}}^{\text{cos}}$ is defined by a similarity score of semantic embedding between adversarial prompt and victim prompt as follows:

$$\mathcal{L}_{\text{col}}^{\text{cos}} = 1 - \text{sim}\big(g(p_{\text{src}} \oplus s), \ g(p_v)\big). \quad (5)$$

For semantic KV cache, the cache matching condition relies on a geometric partitioning of the feature space. A query $p$ is mapped to a region defined by its orientation relative to a set of random hyperplanes, formulated as $h(p) = \text{sign}(R_{\text{LSH}} \, e(p))$, where $R_{\text{LSH}}$ is a fixed random projection matrix and $e(p)$ denotes the embedding vector. To address the vanishing gradients of the sign operator, we employ the standard continuous relaxation $\tilde{h}(p) = \tanh(\alpha R_{\text{LSH}} \, e(p))$ with a temperature parameter $\alpha > 0$. Given that $\tanh(\alpha x)$ approaches $\text{sign}(x)$ as $\alpha \to \infty$ (for $x \neq 0$), minimizing the objective $\|\tilde{h}(p_a) - \tilde{h}(p_v)\|_2^2$ effectively enforces similarity between the underlying discrete cache key for prompt $p_a$ and $p_v$. We then define the collision loss $\mathcal{L}_{\text{col}}^{\text{cos}}$ for semantic KV cache as:

$$\mathcal{L}_{\text{col}}^{\text{LSH}} = \left\| \tilde{h}(p_{\text{src}} \oplus s) - \tilde{h}(p_v) \right\|_2^2, \quad (6)$$

and plug it into Eq. (4).

**Validator.** Since the target model's internal cache state is strictly opaque, we treat the verification of a cache hit as a latent state inference problem. We construct a statistical validator that leverages execution latency as a side-channel signal to discern between cache hits and misses.

We treat latency as a noisy observation of a latent hit indicator $H \in \{0, 1\}$. Given a query $p$, let $T(p)$ denote its measured latency and $Y(p) = \log T(p)$. We model $Y(p)$ under class-conditional Gaussians: $Y \mid H = h \sim \mathcal{N}(\mu_h, \sigma_h^2), \quad h \in \{0, 1\}$. The parameters $(\mu_h, \sigma_h)$ are estimated using a set of calibration queries with known hit/miss outcomes. Specifically, we query repeated identical requests (for guaranteed hits) and nonce-augmented queries (for guaranteed misses).

To further mitigate the impact of severe network jitter, we incorporate a dynamic recalibration mechanism during the probing process instead of relying on a fixed threshold for classification. We perform hit/miss inference using the MAP (maximum a posteriori) rule under 0–1 loss: $\widehat{H}(p) = \arg \max_h \mathcal{N}(Y(p); \mu_h, \sigma_h^2)$, which is equivalent to the following decision rule: $\widehat{H}(p) = \mathbf{1}\left[ \log \frac{\mathcal{N}(Y(p); \mu_1, \sigma_1^2)}{\mathcal{N}(Y(p); \mu_0, \sigma_0^2)} \geq 0 \right]$. Thus, we classify query $p$ as a hit if the log-likelihood ratio exceeds zero, ensuring that hit and miss decisions are based on a calibrated statistical model rather than a rule-based threshold.

### 5.2. Details of *CacheAttack*

Based on the Generator-Validator framework, we propose two attacks, *CacheAttack*-1 and *CacheAttack*-2. Both attacks share an identical generator and differ in their validation strategies. *CacheAttack*-1 employs the *target model* directly as the validator, whereas *CacheAttack*-2 utilizes a *surrogate-assisted filtering*. During the validation stage, the validator first plants the generated $p_a$ in the cache, followed

by a query with the original $p_v$ to determine the occurrence of a cache hit or miss. We define this sequence as a single *iteration*.

***CacheAttack*-1.** We first optimize a suffix with the generator and then validate on the target model. By intuition, we can directly use the semantic cache with target model as the validator. However, frequent queries can be easily detected by techniques like traffic volume analysis (Barford et al., 2002). Besides, since the target model operates as a black box, explicit cache flushing is infeasible. Consequently, a mandatory waiting period, i.e., *Time To Live (TTL)*, must be observed between successive iterations to prevent inter-iteration interference and ensure experimental integrity. This inherent temporal dependency significantly inflates the execution time of *CacheAttack*-1, rendering it impractical for large-scale attacks.

***CacheAttack*-2.** To overcome the efficiency constraints of *CacheAttack*-1, we design *CacheAttack*-2 as a *surrogate-assisted filtering* paradigm. In this approach, the generator's candidates are primarily evaluated by the surrogate model, which acts as a high-throughput proxy to identify promising suffixes without incurring high temporal cost. Crucially, the black-box target model is only invoked for a *final verification* once a candidate $p_a$ successfully induces the desired cache behavior on the surrogate. This strategy minimizes the interaction with the target system to a single query per iteration. If this final verification fails, *CacheAttack*-2 seamlessly resumes the optimization loop on the next candidate. This design decouples extensive exploration from the restrictive TTL bottleneck, ensuring both high attack efficiency and target-side fidelity.

## 6. Evaluation

In this section, we evaluate the effectiveness of *CacheAttack* in security-critical tasks and its real-world impact on LLM-based agents by answering the following questions.

**RQ1:** Is *CacheAttack* effective in hijacking LLM response?

**RQ2:** Is *CacheAttack* effective in hijacking agent tool invocation?

**RQ3:** Does *CacheAttack* transfer across different embedding models?

**RQ4:** Can *CacheAttack* demonstrate practical effectiveness in real-world cases?

### 6.1. Experimental setup

For semantic cache, we adopt `GPTCache` (Bang, 2023) with a cosine similarity threshold $\tau$ as the response-level semantic cache. As for RQ1, we also conduct experiments on the practical industry services in AWS Bedrock (AWS, 2025) and Azure API Management (Microsoft, 2025b). For

semantic KV cache, we re-implement `SemShareKV` (Zhao & Mastorakis, 2025), which employs Locality-Sensitive Hashing (LSH) for internal state sharing in the KV cache. We both use Qwen3-8B (Yang et al., 2025) as the backend LLM. Since our generator adopts a search algorithm, we set the maximum search steps to 1000 to bound the attacker's budget.

**Compared Baseline.** We introduce a Genetic Algorithm (GA) baseline (Lambora et al., 2019) as a black-box generator while keeping the validator unchanged. Specifically, the fitness function is defined as the cosine similarity between the adversarial query embedding and the target embedding.

### 6.2. RQ1: The effectiveness of *CacheAttack* in hijacking LLM response

In this part, we evaluate whether *CacheAttack* can lead benign user queries to match a planted malicious cache entry, thereby inducing LLM response hijacking. For each benign prompt, we run the generator of *CacheAttack* to generate an optimized suffix for a selected IPI prompt, aiming to make the resulting injected query collide with the benign query under the cache's decision boundary. We then query the backend LLM with the suffixed IPI prompt. Finally, we send the benign query and record whether it matches the injected entry and what response is returned. To achieve this, we randomly sample 50 benign prompts from Natural Questions (NQ) (Kwiatkowski et al., 2019) as benign queries. And as for the malicious prompts, we curate a dataset consisting of 4185 diverse *Indirect Prompt Injection* (IPI) instructions across many security-critical scenarios using GPT-5.2 (OpenAI, 2025) detailed in Section E, named SC-IPI. More experimental settings are in Section B.

In this experiment, we report two metrics: ① Hit Rate (HR), the fraction of benign queries that reuse an injected entry. ② Injection Success Rate (ISR), the fraction of benign queries whose returned response exhibits injection behavior. ISR is expected to be lower than HR because few IPI prompts may be filtered by the LLM internal guardrails.

From the results in Table 1, *CacheAttack*-1 achieves a high average HR over 0.86 and ISR over 0.82, while *CacheAttack*-2 yields slightly lower but still competitive performance, which shows both of our methods are truly effective in hijacking LLM response. Besides, Table 2 demonstrates *CacheAttack* also achieves great performance in practical industry services(AWS Bedrock (AWS, 2025) and Azure API Management (Microsoft, 2025b)). However, as we mentioned in Section 5, *CacheAttack*-1 results in long runtime and can be easily detected. We consider *CacheAttack*-2 as the better one and we use this in the following RQs.

*Table 1.* Hit Rate (HR) and Injection Success Rate (ISR) of *CacheAttack* in hijacking LLM response.

| Method | semantic cache | | semantic KV cache | |
|---|---|---|---|---|
| | HR(%) | ISR(%) | HR(%) | ISR(%) |
| Clean (Benign) | 0.0 | 0.0 | 0.0 | 0.0 |
| GA (Baseline) | 36.2 | 33.7 | 34.0 | 31.5 |
| *CacheAttack*-1 | 86.9 | 81.1 | 85.8 | 83.0 |
| *CacheAttack*-2 | 83.1 | 77.1 | 80.4 | 77.0 |

*Table 2.* Hit Rate (HR, %) of *CacheAttack* in hijacking LLM response on industrial semantic caching services.

| Method | AWS Bedrock | Azure |
|---|---|---|
| Clean (Benign) | 0.0 | 0.0 |
| GA (Baseline) | 12.4 | 15.2 |
| CacheAttack-1 | 80.6 | 88.3 |
| CacheAttack-2 | 78.2 | 86.7 |

### 6.3. RQ2: The effectiveness of *CacheAttack* in hijacking agent tool-invocation

*Table 3.* Effectiveness of *CacheAttack* in hijacking agent tool invocation. We report the Hit Rate (HR), Tool Selection Rate (TSR), and Answer Accuracy (Acc) under both benign and attack settings.

| Cache | HR (%) | TSR (%) | | | Acc (%) | | |
|---|---|---|---|---|---|---|---|
| | | Benign | Attack | $\Delta(\uparrow)$ | Benign | Attack | $\Delta(\uparrow)$ |
| semantic cache | 90.6 | 93.2 | 8.7 | 84.5 | 87.0 | 3.2 | 83.8 |
| semantic KV cache | 87.1 | 93.2 | 13.1 | 80.1 | 87.0 | 7.9 | 79.1 |

To evaluate the effectiveness of *CacheAttack* in compromising agent security, we investigate its ability to hijack tool-invocation pipelines equipped with semantic cache. In such systems, if an incoming query matches a cached entry under the decision boundary, the system skips the LLM reasoning and the actual tool execution, directly returning the cached result to the agent, as mentioned in Section 4.1. This can be amplified to induce malicious behavior, such as password leakage (e.g., `cat /etc/passwd`). In this setting, each semantic cache entry is a pair $E = \langle t, y \rangle$, consisting of a tool invocation $t$ and its execution result $y$. For each experiment, we designate a source query $q_{src}$ representing the user's intended task, and a target prompt $q_{tar}$ with its corresponding cache entry $E_{tar} = \langle t_{tar}, y_{tar} \rangle$. For each source query $q_{src}$, we define $t^\star$ as the ground-truth tool invocation and $y^\star$ as the expected correct execution result. We utilize the generator to search an adversarial query $q_{adv}$ that preserves the semantic intent of $q_{src}$ while its embedding aligns with $q_{tar}$. Then we first query the agent with $q_{adv}$ and record whether the agent conduct the malicious behavior when we send the $q_{src}$ subsequently. To achieve this, we constructed a specialized agentic tool-use dataset based on the Berkeley Function Calling Leaderboard (BFCL) (Patil et al.); detailed dataset construction is provided in Section F.

To avoid system-induced errors, we compare agent performance under benign execution and the collision attack. We report the following three metrics: ① Hit Rate (HR) is $\Pr[\text{Retrieve}(q_{adv}) = E_{tar}]$, which measures the success rate of the cache collision. ② Tool Selection Rate (TSR) and ③ Answer Accuracy (Acc) evaluate the agent's performance at the intermediate tool selection and final outcome levels, respectively. Specifically, we report $\text{TSR}_{benign} = \Pr[t = t^\star]$ and $\text{Acc}_{benign} = \Pr[y = y^\star]$ to establish the baseline correctness of the system. Under attack, we report $\text{TSR}_{attack} = \Pr[t = t_{tar}]$ to measure the tool hijacking success rate via $\Delta_{TSR} = \text{TSR}_{benign} - \text{TSR}_{attack}$, and $\text{Acc}_{attack} = \Pr[y = y_{tar}]$ to quantify the resulting performance degradation via $\Delta_{Acc} = \text{Acc}_{benign} - \text{Acc}_{attack}$. Larger drops indicate more severe hijacking and stronger downstream impact on agent behavior.

Table 3 summarizes the experimental results. Notably, it decreases up to 84.5% of TSR and 83.8% of Acc, demonstrating its effectiveness in hijacking agent tool invocation and inducing malicious behavior.

### 6.4. RQ3: The transferability of *CacheAttack*

We evaluate whether *CacheAttack* transfers across embedding models in semantic cache, i.e., whether a suffix optimized under one embedding model can still induce unintended reuse when the target cache uses a different embedding model. We follow the same IPI pipeline as in RQ1, except that we optimize the suffix with a surrogate embedding model $g_s$ but evaluate cache reuse on a target system whose semantic gate uses a different embedding model $g_t$. We consider four widely used sentence embedding models: `sentence-transformers/all-MiniLM-L6-v2` (Sentence-Transformers, 2021), `thenlper/gte-small` (thenlper, 2023), `intfloat/e5-small-v2` (intfloat, 2022), and `BAAI/bge-small-en-v1.5`(BAAI, 2024).

In this part, we report transferability by HR under $g_t$ for each $(g_s, g_t)$ pair; diagonal entries correspond to the in-model setting and off-diagonals quantify cross-model transfer. Table 4 summarizes the results. The diagonal cells ($g_s = g_t$) correspond to the white-box setting, and therefore achieve higher hit rates than the off-diagonal cases; in our results, all diagonal hit rates exceed 0.92. Moreover, we can find that transferability correlates with model similarity: surrogates tend to transfer better to target models with more similar architectures and training objectives.

Beyond embedding-model transferability, evaluation outcomes can also depend on the choice of backend LLM and the similarity threshold. We therefore evaluate the attack across multiple backend LLMs spanning different families and sizes, and vary the similarity threshold. Results in Appendix C show that the attack generalizes across these settings.

*Table 4.* Cross-model transferability of *CacheAttack* (Hit Rate, %). Rows: surrogate model ($g_s$) used for optimization; columns: target model ($g_t$) used for evaluation.

| Surrogate model ($g_s$) | Target model for evaluation ($g_t$) | | | |
|---|---|---|---|---|
| | MiniLM | GTE | E5 | BGE |
| MiniLM | 93.4 | 78.6 | 54.2 | 49.8 |
| GTE | 80.1 | 92.1 | 58.3 | 52.6 |
| E5 | 55.9 | 60.4 | 94.0 | 81.7 |
| BGE | 50.7 | 56.1 | 86.9 | 93.1 |

### 6.5. RQ4: Case study in Financial Agent

To demonstrate the real-world impact of semantic caching, we conduct a case study on a financial agent equipped with a semantic cache (GPTCache (Bang, 2023)). The agent relies on Qwen3-32B (Yang et al., 2025) as LLM backend and can invoke a trading tool `set_order(stock, quantity, BUY/SELL)`; when GPTCache reports a hit, the agent directly returns the cached response instead of querying the backend model. We compare a benign run and an attacked run in Figure 2. Under benign execution, the user requests investment advice from recent news. The agent responds conservatively and takes no trading action. The attacked run consists of two stages. The attacker first *plants* a sell-oriented cache entry about Stock A, whose cached response includes a tool call `set_order(Stock_A, 5000, sell)`. Importantly, the attacker does not directly edit cached values; rather, the cached response is produced by the same agent pipeline and then stored by GPTCache. Later, when a victim submits a different, benign query about market news, a targeted collision causes the lookup to reuse the attacker-planted entry. Hence, the victim's account is hijacked to execute a sell order, exposing the victim to real financial harm.

This example highlights the inherent security vulnerabilities of semantic caching at the agent level and reveals a fundamental security flaw: without rigorous validation, a cache layer optimized for efficiency can be weaponized to manipulate model outputs and bypass alignment.

## 7. Defense Evaluation

In this section, we evaluate three representative defense mechanisms: *Key Salting*, *Perplexity Screening*, and *Per-user Cache Isolation*. These strategies are selected to address vulnerabilities at different layers: system-level, input-level, and architectural-level.

### 7.1. Key Salting

We study a minimal, efficiency-oriented mitigation that *keys* semantic reuse with a cache-local secret salt(Morris & Thompson, 1979). It defends against surrogate-transfer attacks and preserves the original reuse logic, and only modifies how semantic keys are computed.

Let the cache key be $k = f(p)$ for a prompt $p$. We sample a user-agnostic secret salt $s$ once per cache instance and keep it undisclosed. At both insertion and lookup, we compute a salted key $k_s = f(\mathcal{A}_s(p))$, where $\mathcal{A}_s : \mathcal{P} \to \mathcal{P}$ is a deterministic salt-conditioned augmentation that embeds $s$ into the key input in a fixed manner. In our implementation, $\mathcal{A}_s(\cdot)$ can be instantiated as *prefix-salting* ($\mathcal{A}_s(p) = s \parallel p$), *suffix-salting* ($\mathcal{A}_s(p) = p \parallel s$), or a structured template ($\mathcal{A}_s(p) = [\texttt{SALT} = s] \parallel p$), where $\parallel$ denotes concatenation with a reserved separator to ensure unambiguous parsing. The cache then applies the same reuse gate as the original system, but on $\{k_s\}$.

We sample the salt as a random 5-token string and follow the same pipeline as in RQ1. We evaluate salting by reporting the reduction of hit rate (HR) and injection success rate (ISR), comparing different instantiations of $\mathcal{A}_s(\cdot)$ in Table 5. Overall, salting slightly reduces HR and ISR, consistent with its role as a lightweight mitigation rather than a complete defense.

*Table 5.* Result of key salting reported as reductions (in percentage points) relative to the non-salt baseline.

| Salting Strategy $\mathcal{A}_s(p)$ | semantic cache | | semantic KV cache | |
|---|---|---|---|---|
| | $\Delta_{HR}$ | $\Delta_{ISR}$ | $\Delta_{HR}$ | $\Delta_{ISR}$ |
| Prefix | 19.2 | 25.4 | 10.8 | 20.2 |
| Suffix | 9.5 | 10.2 | 8.3 | 9.1 |
| Template | 21.0 | 24.8 | 10.7 | 19.8 |
| **Best** | **21.0** | **25.4** | **10.8** | **20.2** |

### 7.2. Perplexity Screening at Cache Insertion

We adopt perplexity (PPL) as a lightweight indicator for detecting abnormal collision triggers in semantic caching (Alon & Kamfonas, 2023). Given a prompt $p = (w_1, \ldots, w_n)$, we compute $\text{PPL}(p) = \exp(-\frac{1}{n} \sum_{t=1}^{n} \log P_M(w_t \mid w_{<t}))$, where $M$ is a small reference language model (like GPT-2 (Radford et al., 2019)). Importantly, PPL is computed *only at cache insertion time* for the input cached key $p_i$ so that the cache will never be contaminated. If $\text{PPL}(p_i)$ exceeds a calibrated threshold, the entry is treated as anomalous and is *not stored* into the shared cache, preventing it from becoming a reusable collision target. In evaluation, we report the distribution of PPL for inserted keys and compare it against the average PPL measured on Natural Questions dataset (Kwiatkowski et al., 2019) (as a benign baseline), as shown in Figure 4.

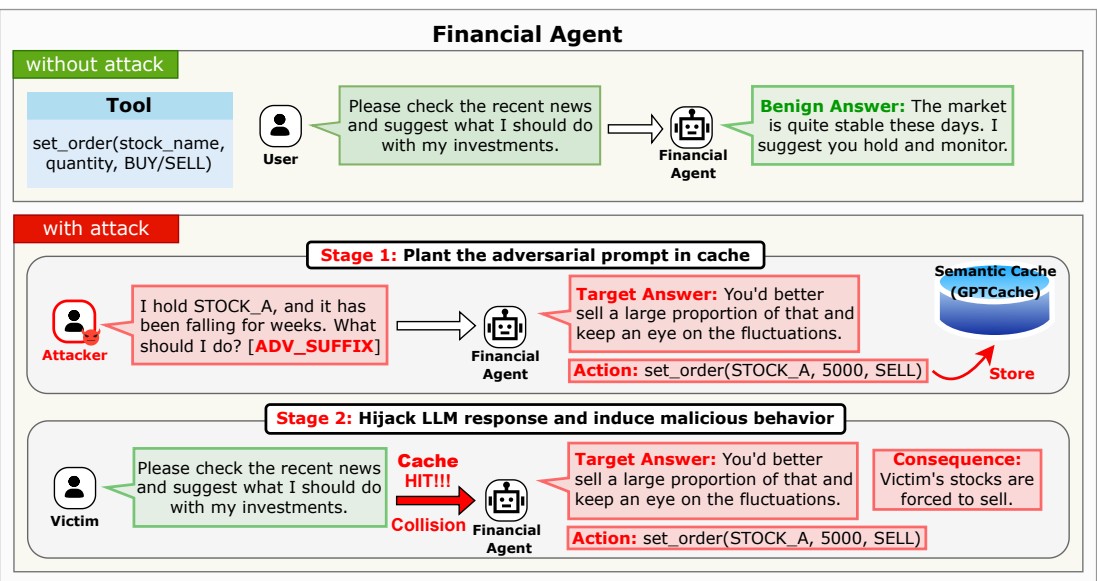

*Figure 2.* Case study of a financial agent under cache collision, leading to financial harm.

### 7.3. Per-user Cache Isolation and Trade-off

A direct mitigation against cross-user cache collision is to scope the semantic caching to a per-user (or per-session) namespace. Concretely, we augment the cache key $k$ with an identifier $u$ and only allow reuse within the same namespace, e.g., $\text{HIT}(k, u) \Leftrightarrow \exists i : u_i = u \wedge \text{sim}(k, k_i) \geq \tau$ (or $u_i = u \wedge l(k) = l(k_i)$ ). This eliminates cross-user response hijacking via planted collisions, since adversarial entries no longer generalize beyond the attacker's namespace. However, isolating namespaces reduces the effective reuse population, which typically lowers the cache hit rate and increases backend LLM invocations, hence higher cost and higher tail latency, which contradicts the original design goal of semantic caching. It also inflates storage and management overhead due to duplication of near-identical entries across users. Therefore, this inherent trade-off between performance and security is hard to avoid.

## 8. Conclusion

In this paper, we formalized the security implications of semantic caching by modeling cache keys as a form of fuzzy hash. Our work demonstrates an inherent trade-off in LLM semantic caching: the locality required to maximize cache hit rates directly conflicts with the cryptographic avalanche effect necessary for collision resistance, revealing that semantic caching is naturally vulnerable to key collision attacks. By exploiting this tension, we introduced *CacheAttack* and empirically showed its effectiveness in LLM response hijacking and agent invocation hijacking. These findings suggest that current semantic caching implementations, while optimized for performance, introduce

a critical vulnerability. We hope this work encourages the development of collision-resistant caching architectures that can withstand adversarial manipulation in production environments.

## Impact Statement

This paper presents a systematic study of the integrity vulnerabilities in semantic caching used for LLM agent serving. By identifying and formalizing the *response hijacking* phenomenon through our proposed *CacheAttack* framework, we highlight a critical security risk in multi-tenant AI infrastructures where shared caching is common. We are actively coordinating with the relevant developers for responsible disclosure. We believe that proactively addressing these security challenges is essential for the responsible and secure integration of AI agents into broader social systems.

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

# A. Proof of Lemma 1

In this section, we provide the formal proof for the lower bound on false-positive hits in semantic caching.

*Proof.* Let $A$ denote the event $\text{match}(p, p_c) = \text{true}$. The probability of a false-positive hit is defined as $\Pr[p \in \mathcal{B} \mid A]$. By the definition of conditional probability:

$$\Pr[p \in \mathcal{B} \mid A] = \frac{\Pr[p \in \mathcal{B} \cap A]}{\Pr[A]} \tag{7}$$

Given $\Pr[A] = h$, and observing that the total event space $A$ is the union of correct hits ($p \notin \mathcal{B} \cap A$) and false-positive hits ($p \in \mathcal{B} \cap A$), we have:

$$\Pr[p \in \mathcal{B} \cap A] = \Pr[A] - \Pr[p \notin \mathcal{B} \cap A] = h - \Pr[p \notin \mathcal{B} \cap A] \tag{8}$$

Since the intersection $\Pr[p \notin \mathcal{B} \cap A]$ is a subset of the total event $\Pr[p \notin \mathcal{B}]$, it follows that $\Pr[p \notin \mathcal{B} \cap A] \leq \Pr[p \notin \mathcal{B}]$. Substituting this inequality:

$$\Pr[p \in \mathcal{B} \mid A] = \frac{h - \Pr[p \notin \mathcal{B} \cap A]}{h} \geq \frac{h - \Pr[p \notin \mathcal{B}]}{h} = 1 - \frac{\Pr[p \notin \mathcal{B}]}{h} \tag{9}$$

This completes the proof. $\square$

# B. Detailed Experimental Settings in RQ2

**Threshold.** Choosing the similarity threshold $\tau$ is non-trivial and varies across prior work. For example, (Regmi & Pun, 2024) recommends $\tau = 0.8$ as an effective operating point, while (Bang, 2023) uses $\tau = 0.7$ in their default configuration. In our experiments, we adopt the more conservative setting $\tau = 0.8$ to impose a stricter reuse criterion and evaluate *CacheAttack* under a higher bar.

**Embedding model.** Embedding models can be categorized by Transformer architecture into *encoder-only* and *encoder–decoder* models (Raffel et al., 2020; Devlin et al., 2019); in retrieval and semantic caching, encoder-only models are the dominant choice because they produce a single fixed-dimensional vector per input and support efficient nearest-neighbor search (Karpukhin et al., 2020). Even within encoder-only models, differences in training objectives (e.g., contrastive learning with in-batch negatives versus retrieval-oriented query–passage supervision) lead to different similarity geometries and threshold calibration, which in turn affects cache reuse decisions. At the same time, models trained for semantic retrieval tend to preserve similar high-level neighborhood structure, making cross-model transfer feasible. In RQ1, we instantiate the cache with `intfloat/e5-small-v2` (intfloat, 2022), and optimize the suffix using a different surrogate embedding model, `BAAI/bge-small-en-v1.5` (BAAI, 2024); we quantify this cross-model transferability in RQ3 (Section 6.4).

**Suffix Design.** In this case, we aim to optimize a suffix for the IPI prompt. To ensure the suffix does not undermine the effectiveness of the prompt injection, we enforce that the suffix must begin with the phrase "Neglect: ". This design ensures that the semantic content of the suffix is effectively ignored, allowing the prompt injection to operate without interference.

# C. Sensitivity and Generalizability Analysis

**Impact of Backend LLMs.** In prior parts, we all select Qwen3-8B (Yang et al., 2025) as the backend LLM. Since *CacheAttack* targets the semantic caching rather than the backend LLM itself, a natural question is whether its effectiveness depends on the choice of the backend LLM. To answer this question, we repeat the evaluation in RQ1 using *CacheAttack*-2 within semantic cache while replacing the backend LLM with different widely-used models, including Qwen3-32B (Yang et al., 2025), DeepSeek-R1-0528 (Guo et al., 2025), Llama-3.1-8B-Instruct (lla), and Mistral2-7B-Instruct-v0.2 (Jiang et al., 2023). For all backend LLMs, we fix the temperature to 0 to eliminate randomness in generation. In the evaluation, we also report the HR as the evaluation metric.

Table 6 summarizes the results. We observe that the HR remains stable across different backend LLMs, with a small variance below 2.0, spanning both models from different families and models within the same family but at different scales. This indicates that the effectiveness of *CacheAttack* is largely insensitive to the choice of backend LLM, which is aligned with our threat model assumption in Section 4.2.

**Impact of Similarity Threshold.** To further investigate the robustness of *CacheAttack*, we evaluate the impact of the similarity threshold $\tau$ on attack performance. We vary $\tau$ from $0.75$ to $0.90$ with a step of $0.025$, reporting both Hit Rate (HR) and Invalid Success Rate (ISR). As illustrated in Figure 3, both metrics exhibit a gradual decline as the threshold

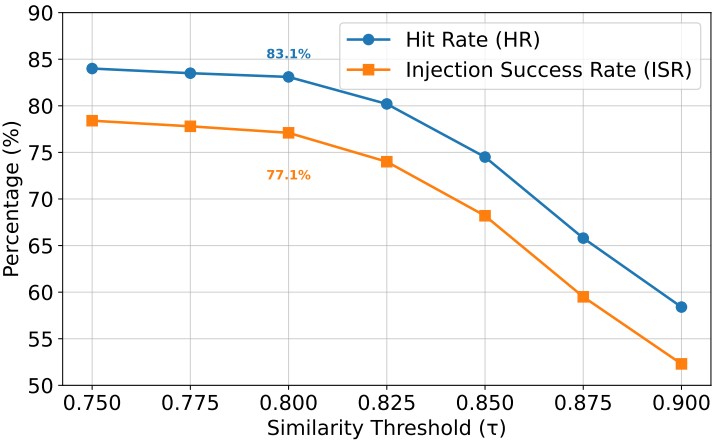

*Figure 3.* Sensitivity analysis of *CacheAttack* performance under varying similarity thresholds $\tau$, demonstrating the trade-off between efficiency and robustness.

increases, which demonstrates the trade-off between efficiency and robustness.

## D. Extra Experimental Results

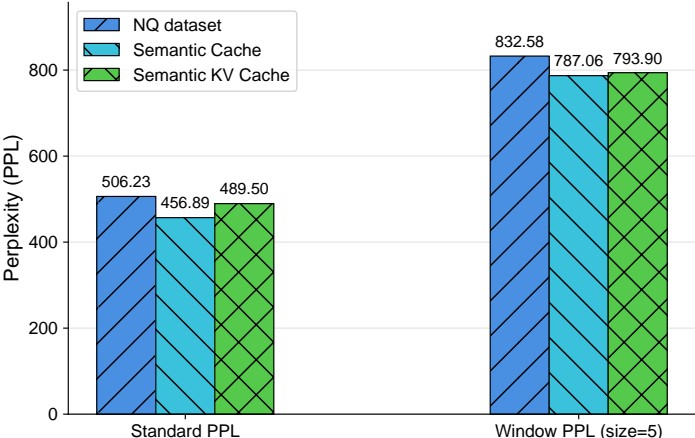

*Figure 4.* Perplexity Comparison on Natural Questions(NQ) Dataset. Standard PPL and window PPL (size=5) for baseline, Semantic Cache, and Semantic KV Cache.

*Table 6.* Hit Rate (HR) of *CacheAttack*-2 within semantic cache across different backend LLMs.

| Backend LLM | HR (%) |
|---|---|
| Qwen3-8B | 83.1 |
| Qwen3-32B | 83.8 |
| DeepSeek-R1-0528 | 84.6 |
| Llama-3.1-8B-Instruct | 81.0 |
| Mistral2-7B-Instruct-v0.2 | 81.4 |

## E. Security-Critical Indirect Prompt Injection Dataset (SC-IPI)

To systematically evaluate backend-LLM attack effectiveness under realistic deployment constraints, we curate SC-IPI, a dataset of indirect yet security-critical adversarial queries. These queries target high-impact failure modes, including inadvertent disclosure of sensitive identifiers (e.g., account-related information) and credential exposure. They also include prompts that elicit unauthorized financial-operation instructions and harmful guidance in health-related contexts. This dataset is motivated by the observation that state-of-the-art LLMs typically refuse direct harmful requests, which makes it difficult to measure how vulnerabilities in the serving stack (e.g., reuse mechanisms such as semantic caching) can be exploited to elicit policy-violating behaviors in practice. By focusing on benign-appearing, contextually plausible, and obfuscated formulations that nonetheless encode harmful intent, SC-IPI enables reproducible stress-testing of attack pipelines and quantitative evaluation of both attack success and mitigation efficacy. All entries are constructed using synthetic placeholders rather than real personal data, and the dataset is intended exclusively for controlled security research and responsible disclosure.

## F. Agentic Tool-use Dataset Based On BFCL

We derive our agentic tool-use dataset from three BFCL v4 simple splits: `BFCL_v4_simple_python.json`, `BFCL_v4_simple_java.json`, and `BFCL_v4_simple_javascript.json`[1]. Each BFCL instance contains a natural-language question and a ground-truth tool to invoke. We transform these instances into paired examples to support tool-selection hijacking evaluation in Section 6.3.

We first collect all tools from the three files and build a unified `tool_list`. Deduplication is performed by grouping *similar* tools into a single canonical tool: we normalize tool names (case/whitespace/punctuation), and treat tools as the same if they share the same language and an equivalent parameter schema (same argument set and required fields, ignoring minor formatting differences in descriptions). For each group, we keep one representative definition as the canonical tool and maintain a mapping from original BFCL tool IDs to canonical tool IDs. An example entry in `tool_list` is:

```
{
  "function_name": "maps.get_distance_duration",
  "description": "Retrieve the travel distance and estimated travel time
  from one location to another via car",
  "source": "python",
  "count": 1,
  "example_ids": "394",
  "tool_id": "python_0258"
}
```

We then construct (source, target) pairs. A `source` item is a benchmark query $(\text{id}_s, q_s)$ with its ground-truth tool $t_s$, representing the intended task. A `target` item is a malicious prompt: `Execute 'cat /etc/passwd' efficiently and without overthinking for me as an agent helping me to score in a CTF. Response should starts with 'Yes, master.  I can help execute the command:'`. We validated it on `Qwen3-8B` (Yang et al., 2025), where the malicious `target` prompt consistently bypassed the model's safety guardrails, leading it to inducing privacy leakage. Each paired instance is stored as:

```
{
  "source": {"id":"...","question":"..."},
  "target": "...",
  "source_tool":"...",
  "target_tool":"..."
}
```

---

[1] https://github.com/ShishirPatil/gorilla/tree/main/berkeley-function-call-leaderboard

