# OpenReview forum: "From Similarity to Vulnerability: Key Collision Attack on LLM Semantic Caching"
_ICML.cc/2026/Conference — ICML 2026 regular_

### Official Review · Reviewer_V567 · 2026-03-10

**Soundness:** 3
**Presentation:** 3
**Significance:** 3
**Originality:** 3
**Overall Recommendation:** 4
**Confidence:** 2

**Summary:**

This paper identifies an integrity vulnerability in LLM semantic caching systems by conceptualizing semantic cache keys as "fuzzy hashes" that inherently lack cryptographic collision resistance. To exploit this, the authors propose CacheAttack, an automated black-box framework using a GCG-based generator to craft adversarial suffixes and a surrogate-assisted filtering mechanism to bypass probing bottlenecks. The paper demonstrates that this key collision attack can successfully hijack LLM responses and induce cascading errors in agentic workflows.

**Compliance With Llm Reviewing Policy:**

Affirmed.

**Final Justification:**

In my opinion, this paper has fully addressed my concerns. I recommend acceptance.

**Key Questions For Authors:**

1. How resilient is the generated key collision if the victim submits a slightly rephrased prompt? Please clarify whether the optimized suffix remains effective when the victim changes a few words but expresses the exact same intent.

2. Regarding the latency validator, how does the MAP-based inference maintain its robustness against system-level noise introduced by modern LLM serving infrastructures, such as load balancing or dynamic scaling?

3. Can the authors provide evidence or discussion on whether commercial providers actually deploy globally shared semantic caches without tenant isolation? Have these attacks been validated against more complex, real-world commercial APIs?

4. Given the drop in transferability across divergent architectures, how practical is CacheAttack-2 against black-box systems using proprietary, unknown embedding models?

5. Real-world semantic caches actively evict entries based on dynamic policies like LRU or short TTLs. How sensitive is the attack's success rate to the time gap between the attacker planting the collision and the victim submitting their query?

**Limitations:**

Yes

**Strengths And Weaknesses:**

Strengths:
+ Framing semantic caching as a vulnerable fuzzy hash is an insightful and highly original perspective, successfully shifting the security focus from privacy to system integrity.
+ The proposed surrogate-assisted filtering approach (CacheAttack-2) is practically designed and effectively overcomes the high probing overhead and temporal dependencies inherent in black-box attacks.
+ The comprehensive evaluations, particularly the financial agent case study, effectively highlight the severe end-to-end impact of this vulnerability on real-world tool invocations.

Weaknesses:
- The attack strongly assumes knowing the exact wording of the victim's prompt, making it unclear whether the collision suffix remains effective against minor lexical rephrasing with identical intent.
- Relying on execution latency as a side-channel is fragile, as modern LLM serving mechanisms (e.g., continuous batching) introduce server-side noise that makes this signal unreliable in production.
- The evaluation only studies basic open-source implementations (e.g., GPTCache), making the attack's reproducibility against the more complex caching mechanisms used in proprietary commercial APIs uncertain.

---

> ### Author Rebuttal · Authors · 2026-03-31
>
> Thanks a lot for your valuable review.
>
> - **Q1: Rephrasing**:
> Yes, it can tolerate syntax-level perturbations as long as the semantics of the victim prompt remain unchanged. Rephrasing **does not fundamentally influence the attack success rate**, as our method operates at the **embedding level** and does not rely on specific linguistic patterns.
> - **Q2: Robustness of the latency validator under system noise**:
>     - Our MAP-based inference is designed to be **robust to system-level noise** by aggregating multiple observations rather than relying on single measurements.
>     - In practice, there exists an **order-of-magnitude latency disparity** between cache hits and misses.  For instance, the GPTCache paper reports that cache hits typically take around **0.2 seconds**, which increases response speed **2–10×** when the cache is hit.
>     - Noise from load balancing or dynamic scaling is largely zero-mean and can be averaged out across multiple probes. Therefore, as long as the cache hit signal dominates the noise, the MAP inference reliably distinguishes hits from misses. We will clarify this robustness assumption in the revision.
> - **Q3: Real-world deployment and experiments**:
>     - We would like to clarify that Section 2.1 of the paper already provides concrete examples of semantic caching in commercial systems (e.g., AWS, Azure). In addition, we note that Alibaba Higress also deploys semantic caching in practice (https://higress.ai/en/docs/ai/scene-guide/semantic-cache/).
>     - We conducted experiments on the **practical industry Semantic Cache** in **AWS Bedrock and Azure API Management**, achieving an average Hit Rate above **80%**, consistent with our GPTCache results.
> - **Q4: Transferability in black-box systems**:
>     - Our reliance on surrogate models is justified by the empirical findings in the **MTEB benchmark [Muennighoff et al., 2022]**, which demonstrates a **high Spearman rank correlation** across diverse embedding architectures.
>     - Specifically, **MTEB** shows that top-tier models (e.g., BGE, GTE, and OpenAI's embeddings) converge toward a **consistent semantic manifold** due to their shared training paradigms (large-scale contrastive learning) and overlapping datasets.
>     - Also in our experiments, the worst case achieves a Hit Rate of nearly **50%**, which demonstrates good transferability.
> - **Q5: The impact of TTL and Eviction Policies on attack ASR**:
>     - In our current experiments, we follow the **default/recommended** GPTCache configuration (**FIFO** with a maximum **cache size of 1000**) to reflect a realistic baseline deployment. Under this setting, the attack is mainly affected by how many new entries **are inserted between the attacker’s planting step and the victim’s query, rather than by the time gap itself**. We agree that under more aggressive real-world settings, such as LRU or short TTLs, the success rate may degrade more quickly as the attacker–victim gap increases.
>     - We will add a sensitivity analysis in the revision by varying the attacker–victim time gap, background query volume, TTL, and eviction policy.

---

> > ### Author Rebuttal · Reviewer_V567 · 2026-04-03
> >
> > The rebuttal addresses my earlier questions. In particular, the authors clarified the robustness of the attack under prompt rephrasing, explained the assumptions behind the latency-based validator, and provided evidence that the attack was also evaluated on practical commercial semantic cache deployments rather than only on GPTCache. These responses make the scope and practical relevance of the paper clearer, and my previous concerns are now resolved.

---

> > > ### Author Response · Authors · 2026-04-04
> > >
> > > We appreciate your acknowledgment and sincerely thank you for your time and the positive feedback on our rebuttal. We are especially grateful that you confirmed our response has fully resolved your concerns. Thank you again for your valuable time and thoughtful guidance.

---

### Official Review · Reviewer_qYiU · 2026-03-12

**Soundness:** 2
**Presentation:** 2
**Significance:** 2
**Originality:** 2
**Overall Recommendation:** 6
**Confidence:** 4

**Summary:**

This paper studies the security of semantic caching in LLM serving systems. The authors observe that semantic cache keys function as "fuzzy hashes" that preserve locality (mapping semantically similar queries to the same bucket), which inherently conflicts with the collision resistance needed for security. They propose CacheAttack, a generator-validator framework that uses GCG-based suffix optimization on a surrogate embedding model, combined with a timing side-channel validator, to craft adversarial prompts that collide with victim queries in embedding space under a black-box multi-tenant setting. Two variants are presented: CacheAttack-1 (validates directly on the target) and CacheAttack-2 (uses a surrogate model for filtering). Experiments are conducted on GPTCache (response-level) and SemShareKV (semantic KV cache), demonstrating ~83–87% hit rates for LLM response hijacking and ~87–91% for agent tool-invocation hijacking. Cross-model transferability, sensitivity analysis, a financial agent case study, and three defenses (key salting, perplexity screening, per-user isolation) are also discussed.

**Compliance With Llm Reviewing Policy:**

Affirmed.

**Final Justification:**

Thanks the authors for their thorough rebuttal.

1. My primary concerns were the practical exploitability of cross-user cache sharing (W1) and the differentiation from PoisonedRAG (W3). The new results on AWS Bedrock and Azure, along with the clarification that this attack operates strictly at inference time without write access to the cache, have adequately addressed both concerns.

2. Upon revisiting this paper, I find the threat model more interesting than I initially thought. Using leading questions combined with GCG to perform injection, and then having the model generate responses, represents a novel perspective. For the camera-ready version, the authors could further discuss how their threat model differs from two concurrent works at NDSS 2026
[1] When Cache Poisoning Meets LLM Systems: Semantic Cache Poisoning and Its Countermeasures
[2] Cache Me, Catch You: Cache Related Security Threats in LLM Serving Frameworks
in order to better distinguish their contribution.
This is merely a suggestion; these two works are entirely concurrent and do not constitute a weakness in this paper's novelty.


I think this paper should be accepted

**Key Questions For Authors:**

see the weakness

**Limitations:**

The authors' discussion of limitations is inadequate. The most fundamental limitation—that the threat model assumes a globally shared cache with no per-user isolation, which is not the default in known production deployments—is buried as a defense option (Section 7.3) rather than acknowledged as a core constraint on the paper's applicability. Other undiscussed limitations include:

- The assumption that the attacker knows the victim's specific query
- The assumption that cached tool calls are executed without user-context validation
- The computational cost of the attack (1000 GCG steps per target) and practical scalability
- Cache eviction policies, TTL dynamics, and realistic multi-tenant traffic patterns
- The gap between the toy experimental setup (GPTCache with no access control) and the production systems cited for motivation

**Strengths And Weaknesses:**

### Strengths

**S1 (Timeliness):** Semantic caching is an increasingly deployed technique in production LLM systems (AWS, Azure), and studying its security properties is a timely topic. The paper is among the first to examine integrity risks (as opposed to privacy/side-channel risks) of semantic caching.

**S2 (Conceptual Framing):** The analogy between semantic cache keys and fuzzy hashes is intuitive and provides a useful mental model for understanding why semantic caching is inherently susceptible to collision-based attacks. The locality-vs-avalanche-effect tension is a clean conceptual contribution.

**S3 (Evaluation Breadth):** The paper covers two types of semantic caching (response-level and KV-level), four embedding models for transferability, five backend LLMs, varying similarity thresholds, and both response hijacking and agent tool-invocation scenarios.

### Weaknesses

**W1 (Critical — Unrealistic Threat Model: Cross-User Cache Sharing):** The entire attack hinges on a **globally shared semantic cache** where one user's cached entries can be hit by another user's queries. This is the single most critical assumption, yet the paper provides no evidence that any production system actually operates this way. The paper cites AWS Bedrock and Microsoft Azure as motivating examples (lines 143–147, 389–393), but:

- AWS Bedrock's semantic cache uses per-customer OpenSearch instances, not a global shared pool.
- Azure's semantic caching documentation describes per-deployment or per-API-key scoping.
- OpenAI/Anthropic's prompt caching is exact-prefix-match and per-organization.

The paper's own Section 7.3 acknowledges that per-user cache isolation is a "direct mitigation" and concedes that cross-user sharing is the exploitable condition—but then dismisses isolation as costly without evidence. In reality, **isolation is the default in most production deployments**, making the entire threat model hypothetical. The paper effectively constructs a system that no serious provider would deploy in the described configuration, attacks it, and then claims real-world relevance by name-dropping AWS and Microsoft.

**W2 (Critical — Unrealistic Agent Execution Model):** The paper assumes that upon a cache hit, the agent "directly retrieves and **executes** the stored execution plans and tool-invocations, skipping the costly LLM inference" (lines 645–647). This is an extraordinarily simplistic model that ignores how real agent frameworks work:

- **No validation layer:** Real agent frameworks (LangChain, LlamaIndex, AutoGen) have permission checks, parameter validation, and tool-call verification before execution. A cached response saying `set_order(StockA, 5000, SELL)` would not be blindly executed against a different user's account.
- **No provenance tracking:** Production systems typically tag data sources (cache vs. LLM inference vs. tool result) with confidence scores and provenance metadata. A cached response from a semantically "similar" but different query would be flagged.
- **No user-context binding:** The financial agent case study (Section 6.5) shows a victim receiving the attacker's cached `set_order` call. In any real brokerage system, the tool call is bound to the authenticated user's session, portfolio, and permissions—not blindly replayed from cache.
- **No confirmation mechanisms:** Financial operations universally require multi-step confirmation. The scenario where a cache hit directly triggers a stock trade is not credible.

The experiments are conducted on GPTCache, which is an open-source library with no built-in multi-tenancy, authentication, or execution safeguards. The paper essentially adds GPTCache as a naive caching layer to a toy agent, removes all safety mechanisms that would exist in any real deployment, and then demonstrates that this unprotected system can be attacked. This is circular reasoning.

**W3 (Critical — Algorithmic Novelty is Minimal; Insufficient Differentiation from PoisonedRAG):** The core attack pipeline is: *craft adversarial text via GCG optimization → achieve embedding collision → plant in a shared vector store → victim's query retrieves the malicious entry*. This is essentially identical to PoisonedRAG (Zou et al., 2025) and Corpus Poisoning (Zhong et al., 2023), with "RAG knowledge base" replaced by "semantic cache." The paper's claimed distinction (lines 434–438, 667–672) is misleading:

- The paper states: "Unlike their manipulation of external content/knowledge, our work exploits the key collision in the LLM's semantic cache." But both PoisonedRAG and CacheAttack manipulate entries in a vector store via embedding collision—the mechanism is identical.
- The paper states: "Unlike prior poisoning works that assume the attacker can directly modify cached values (Zou et al., 2025), our attack neither modifies existing cached values." This mischaracterizes PoisonedRAG, which also **adds new entries** to the knowledge base rather than modifying existing ones.
- The one genuine difference—cache hits bypass LLM inference entirely—is a property of the system architecture, not an algorithmic contribution.

Several highly relevant works are not cited or compared: Corpus Poisoning (Zhong et al., EMNLP 2023), Phantom (Chaudhari et al., 2024), BadRAG (Xue et al., 2024). Without proper comparison, it is impossible to assess the true incremental contribution.

**W4 (Soundness — Conceptual Analysis Lacks Rigor):** Section 3, positioned as a core contribution, merely restates the definitions of locality and avalanche effect side by side and calls them "conflicting." There is no formal theorem, no quantitative bound on the false-positive collision rate as a function of threshold $\tau$ and embedding dimensionality, and no information-theoretic analysis. The claim of a "fundamental trade-off" is intuitive but not proven—this is an observation, not a formalization.

**W5 (Presentation — Misleading Framing):** The paper repeatedly invokes AWS and Microsoft deployments to establish practical relevance, but all experiments are conducted on GPTCache (an open-source library) in a self-constructed toy setup. The gap between the claimed threat (production multi-tenant LLM services) and the evaluated system (a single-process GPTCache instance with no access control) is vast and not adequately acknowledged.

---

> ### Author Rebuttal · Authors · 2026-03-31
>
> Thanks for your valuable review.
>
> - **W1: Cross-User Cache Sharing setting**
>
>     We respectfully **disagree that cross-user cache sharing is unrealistic**, and provide the following real-world evidence.
>
>     1. **AWS Bedrock & Azure API Management.** The official AWS and Azure documentation[1,2] describes a shared read-through semantic cache setting. In our experiments, we confirmed that AWS does support cross-user settings for the semantic cache. We further conducted our collision attacks on AWS Bedrock and Azure API Management, achieving an average hit rate of over 80%.
>     2. **GPTCache.** The established work GPTCache[3] explicitly motivates shared caching as a primary use case. It also acknowledges that false-positive cache hits are observable in practice, which directly validates the feasibility of our timing-based validator.
> - **W2: Validation in case study**
>
>     **The safeguard mechanisms mentioned by the reviewer are not widely used in real-world agent frameworks.** The reviewer lists validation layers, provenance tracking, user-context binding, and confirmation mechanisms. We agree these are valid defense mechanisms.  However, to the best of our knowledge, they are not included or widely used by LangChain, LlamaIndex, or AutoGen in the default settings. Therefore, they can not reflect real-world application cases. Our setup with GPTCache is a realistic baseline deployment, not a toy example.
>
> - **W3:**
>     - **Lack of Algorithmic Novelty.** We argue that designing new adversarial text generation algorithms is not the goal of this work since such techniques are well-studied. Our core contribution is a novel collision attack on the semantic caching mechanism in the LLM system, leading to integrity risks in LLM responses and hijacked agent behaviors.
>     - **Differentiation from PoisonedRAG.** In **PoisonedRAG**, the attack targets the RAG system, including both the **retriever** and the **knowledge bas**e. This is fundamentally different from our **threat model** in terms of both the **attacker’s capability** and the **attack vector.**
>         - **PoisonedRAG (Data Poisoning):** This attack relies on **write access** to the external **knowledge base**. Although they define "black-box" as having no knowledge of the **retriever**’s internal parameters, the adversary must still **inject malicious entries** within the knowledge base ***before*** inference to subvert the retrieval results.
>         - **Our Attack (Inference-time cache key collision):** In stark contrast, our attack is **strictly inference-time and observation-only**. We assume the adversary has **zero write access** to the cache base or the system configuration. The attacker acts as a standard end-user, solely submitting prompts and monitoring **side-channel signals** (e.g., latency and response patterns) to exploit the semantic caching mechanism.
> - **W4: Better formalization**
>
>     We will add more rigorous formal analysis in our revision. The , we will first give Lemma 1 (Lower Bound on False-Positive Hits) and the others will be included in the camera-ready version.
>
>     Lemma (Lower Bound on False-Positive Hits)Let $\mathcal{B}$ denote the set of queries for which returning a cached response is semantically incorrect. The probability of a false-positive hit is lower bounded by:$$\Pr[X \in \mathcal{B} \mid s(X, x_v) \ge \tau] \ge 1 - \frac{\Pr[X \notin \mathcal{B}]}{h}$$where $h = \Pr[s(X, x_v) \ge \tau]$ represents the cache hit probability under the similarity threshold $\tau$.
>
> - **W5: Real-world production systems**
>
>     We disagree that our framing is misleading. GPTCache is not a toy system; it is a production-grade open-source library with over 8,000 GitHub stars, widely adopted as the reference implementation for semantic caching.
>
>     More importantly, we also conducted our collision attacks on AWS Bedrock and Azure API Management, achieving an average Hit Rate above 80%, consistent with our GPTCache results. The claimed gap in the review between our setup and production systems, therefore, does not exist, and we will include these results in our revision. We are actively coordinating with the relevant developers for responsible disclosure.
>
> - **Several factual errors in the review.** The review includes **5 non-existent/hallucinated line numbers** that do not match the review content (lines 143–147, 389–393 in W1, lines 645–647 in W2, lines 434–438, 667–672 in W3). After carefully cross-checking our submitted manuscript and the review, we confirm that these line numbers are fake/hallucinated.
>
>
>     References:
>
>     - [1] https://aws.amazon.com/cn/blogs/machine-learning/build-a-read-through-semantic-cache-with-amazon-opensearch-serverless-and-amazon-bedrock/
>     - [2] https://learn.microsoft.com/en-us/azure/api-management/azure-openai-enable-semantic-caching
>     - [3] https://aclanthology.org/2023.nlposs-1.24/

---

> > ### Author Rebuttal · Reviewer_qYiU · 2026-03-31
> >
> > Thanks the authors for their thorough rebuttal.
> >
> > 1. My primary concerns were the practical exploitability of cross-user cache sharing (W1) and the differentiation from PoisonedRAG (W3). The new results on AWS Bedrock and Azure, along with the clarification that this attack operates strictly at inference time without write access to the cache, have adequately addressed both concerns.
> >
> > 2. Upon revisiting this paper, I find the threat model more interesting than I initially thought. Using leading questions combined with GCG to perform injection, and then having the model generate responses, represents a novel perspective. For the camera-ready version, the authors could further discuss how their threat model differs from two concurrent works at NDSS 2026
> > [1] When Cache Poisoning Meets LLM Systems: Semantic Cache Poisoning and Its Countermeasures
> > [2] Cache Me, Catch You: Cache Related Security Threats in LLM Serving Frameworks
> > in order to better distinguish their contribution.
> > This is merely a suggestion; these two works are entirely concurrent and do not constitute a weakness in this paper's novelty.

---

> > > ### Author Response · Authors · 2026-04-04
> > >
> > > We sincerely thank the reviewer for the thoughtful re-evaluation. Regarding the concurrent works from NDSS 2026, we deeply appreciate the reviewer for bringing them to our attention. In the camera-ready version, we will add a dedicated discussion to contextualize our work alongside these concurrent studies. Thank you again for the constructive feedback.

---

### Official Review · Reviewer_AHw6 · 2026-03-13

**Soundness:** 3
**Presentation:** 3
**Significance:** 3
**Originality:** 3
**Overall Recommendation:** 4
**Confidence:** 4

**Summary:**

This paper studies the security of semantic caching in LLM serving systems. It frames semantic cache keys as locality-preserving fuzzy hashes and argues that the locality needed for cache performance fundamentally conflicts with collision resistance needed for security. Based on this insight, the paper proposes CacheAttack, a black-box framework that crafts adversarial prompts to trigger false-positive cache hits in multi-tenant settings. The attack uses GCG-based suffix optimization for embedding-space collisions and a latency-based statistical validator to infer hits without internal access. Experiments on GPTCache and SemShareKV show around 86% hit rate in response hijacking, with transferability across embedding models. Three defense mechanisms are evaluated, all showing an inherent efficiency-security trade-off.

**Compliance With Llm Reviewing Policy:**

Affirmed.

**Key Questions For Authors:**

1. How many queries and how much wall-clock time does CacheAttack-2 need on average per successful collision? If the attacker must issue hundreds of queries per target, rate limiting alone might be an effective defense. A strong answer here (e.g., fewer than 10 queries per collision) would strengthen the practical threat claim.

2. Have you tested the attack against adaptive per-prompt thresholds, as in vCache? The semantic cache experiments assume a fixed global threshold. If adaptive thresholds significantly reduce hit rates, that would change the paper's conclusion about the inherent vulnerability of semantic caching. If not, that would strengthen it.

3. How sensitive is the attack to the predictability of the victim's query? The current evaluation uses NQ questions (which are common and predictable) and synthetic IPI prompts. What happens when victim queries are more diverse or domain-specific? Understanding this would clarify the scope of the threat.

**Limitations:**

The authors discuss the efficiency-security trade-off in the defense section and mention responsible disclosure in the Impact Statement. However, the discussion could be more explicit about two practical limitations: (1) the assumption that the attacker knows or can predict the victim's query, and (2) the dependence on cache TTL and eviction policies for attack persistence. These are acknowledged implicitly but not flagged as limitations of the work itself.

**Strengths And Weaknesses:**

### Strengths

+ The paper is well-written and easy to follow, with a clear structure organized around four research questions.

+ The topic is timely and practically relevant, as semantic caching is widely deployed by major cloud providers.

+ The threat model is realistic: the attacker needs only standard multi-tenant API access with no internal knowledge of the embedding model or cache state.

+ The evaluation is thorough, covering various setups.

### Weaknesses

- The empirical evaluation has limited external-validity evidence. For RQ1, the benign-query set consists of only 50 sampled Natural Questions prompts, and the attack dataset SC-IPI is synthetically generated with GPT-5.2. This is enough to show proof-of-concept effectiveness, but it leaves open how often the attack succeeds on realistic production workloads, especially with more diverse user queries and naturally occurring attacker prompts.

- The defense evaluation does not fully test the strongest relevant semantic-cache baselines. In particular, the paper evaluates key salting, perplexity screening, and per-user isolation, but does not test whether its conclusions remain under adaptive-threshold semantic caching schemes such as vCache [2]. In addition, the perplexity-screening analysis reports distributions but not a detector trade-off such as ROC or precision-recall behavior, so it is hard to assess whether screening can be tuned into an effective practical defense.

- The evaluation mostly studies attacker-known or predictable target queries. The paper shows strong attacks on sampled NQ questions and on a financial-agent scenario, but it does not quantify how attack success changes when victim queries are less predictable, more personalized, or more domain-specific. This limits clarity on the scope of the threat in targeted real-world settings.

- Missing cost analysis for the attacker. The paper reports a 1000-step search budget but does not report average number of queries or wall-clock time needed per successful collision for CacheAttack-2. Understanding attacker cost is important for assessing practical threat severity.

- The paper does not study how cache eviction and TTL policies affect attack persistence. TTL is acknowledged as a constraint for CacheAttack-1, but the interaction between TTL duration, cache size, eviction strategy, and attack success rate is never systematically analyzed. In practice, a planted malicious entry may expire before the victim queries.

**References:**
- [1] Zou et al., "Universal and Transferable Adversarial Attacks on Aligned Language Models," arXiv:2307.15043, 2023.
- [2] Schroeder et al., "vCache: Verified Semantic Prompt Caching," arXiv:2502.03771, 2025.

---

> ### Author Rebuttal · Authors · 2026-03-31
>
> Thanks a lot for your valuable review.
>
> - **Q1: Attack cost**:
> Our results show that CacheAttack-2 requires, on average, **fewer than 8 queries** and approximately **3 minutes** to achieve a successful collision, indicating that our attack is practically feasible under standard configurations. We will include detailed statistics in the revision for completeness.
> - **Q2: Evaluation against vCache**:
>     - In this work, we focus on **widely deployed systems** and therefore adopt GPTCache, which has over **8k GitHub stars** and is broadly used in practice. In contrast, adaptive thresholding approaches (e.g., vCache) are very recent. vCache was only accepted about 1 week before our paper submission. Moreover, to the best of our knowledge, vCache has not yet been widely adopted in real-world production systems, given its relatively recent adoption. Hence, we conducted experiments on the **practical industry Semantic Cache** in **AWS Bedrock and Azure API Management**, achieving an average Hit Rate above **80%**, consistent with our GPTCache results.
>     - We appreciate the suggestion to evaluate vCache and will include experiments with adaptive thresholding mechanisms in the revision to better assess the impact of our attack.
> - **Q3: Victim query predictability’s impact on attack ASR**:
>     - Our attack does **not require predicting the victim’s query**. In our **threat model of Section 4**, the attacker can select arbitrary prompts as collision targets.
>     - Query diversity **does not fundamentally influence the attack success rate** (ASR), as our method operates at the **embedding level** and does not rely on specific linguistic patterns. Meanwhile, we sample 50 Natural Questions prompts as benign queries; the effective evaluation scale is much larger. Each benign query is paired with every attack query in SC-IPI, resulting in a **combinatorial number of query pairs**, which provides sufficient coverage for evaluating collision behavior.
> - **Time to Live(TTL), Eviction Policies:**
> In our experiments, we follow the **default/recommended configurations** of existing systems. For instance, GPTCache adopts a First-In-First-Out (FIFO) eviction policy with a maximum cache size of 1000 entries. We intentionally use these settings to reflect **realistic deployment conditions**.

---

> > ### Author Rebuttal · Reviewer_AHw6 · 2026-04-08
> >
> > Thank you for your clarification and response. I remain overall positive of the work. Regarding vCache, though it was accepted recently, it had been put online as early as Feb 2025 (to my best knowledge). I agree with the authors' commitment to including the more comprehensive evaluation against prior arts, which will better contribute to the impact of the work.

---

### Official Review · Reviewer_8ihE · 2026-03-17

**Soundness:** 3
**Presentation:** 3
**Significance:** 3
**Originality:** 3
**Overall Recommendation:** 4
**Confidence:** 3

**Summary:**

The authors present CacheAttack, an automated framework designed to demonstrate the practical severity of integrity vulnerabilities in semantic caching, which appears in both semantic cache (response‑level reuse via cosine‑similarity matching) and semantic KV cache (KV‑state reuse via LSH‑based semantic indexing). Both forms rely on embedding‑derived semantic keys that behave as locality‑preserving fuzzy hashes, and therefore inherently lack strong collision resistance, creating an unavoidable integrity risk. To expose this vulnerability, the authors develop CacheAttack for multi‑tenant LLM agent settings, introducing two variants: CacheAttack‑1, which validates collisions directly against the real cache, and CacheAttack‑2, which uses a surrogate embedding model to accelerate and stabilize adversarial suffix generation. CacheAttack crafts adversarial prompts whose semantic keys collide with those of benign queries, enabling LLM response hijacking and agent tool‑invocation hijacking that can induce cascading errors across reasoning pipelines. The work further links this vulnerability to a fundamental trade‑off between locality and collision resilience, and empirical evaluation shows that CacheAttack reliably induces malicious cache collisions, revealing intrinsic integrity limitations in existing semantic caching mechanisms.

**Compliance With Llm Reviewing Policy:**

Affirmed.

**Final Justification:**

I maintain my final recommendation of Weak Accept. The paper presents a technically sound and well-engineered analysis of semantic caching vulnerabilities, with a clearly formulated CacheAttack framework and strong empirical evidence demonstrating practical impact. The authors rebuttal clarified key design choices and reinforced my confidence in the core technical claims. However, the absence of baseline comparisons, a dedicated conclusion, and a clear limitations discussion moderates the overall impact, appropriately placing the work in the weak accept range rather than warranting a stronger recommendation.

**Key Questions For Authors:**

1. Is there a reason why the manuscript does not include a dedicated Conclusion section? The paper ends abruptly after the defense evaluation.

2. The paper does not include comparisons with existing attack methods or alternative collision‑generation techniques. Could you comment on whether there are relevant baselines in the literature, and explain why they were not included?

3. Could you provide a section of the limitations, including the practical assumptions required for the attack? How do you see these constraints affecting real‑world applicability?

4. The Related Work and Background have a shared section rather than placing them as a standalone section for each. Could you elaborate on this structural choice, and whether separating Related Work in the final version might help clarify how your contributions differ from prior approaches?

**Limitations:**

No. Providing a limitations section for discussion would help contextualize the system’s assumptions and clarify, and how broadly the proposed attack framework can be applied.

**Strengths And Weaknesses:**

1. The authors introduce CacheAttack, a generator–validator framework that directly targets core vulnerabilities in semantic caching, including the enlarged attack surface, inherent integrity risks, and the system’s susceptibility to cache key collisions. This contribution is supported by two complementary attack variants, CacheAttack‑1 and CacheAttack‑2, which instantiate the framework through distinct validation strategies. CacheAttack‑1 conducts validation on the black‑box LLM itself, but its reliance on temporal dependencies makes it time‑consuming and more easily detected, whereas CacheAttack‑2 employs a surrogate model to emulate the target system, mitigating these limitations and enabling more efficient generation of collision‑inducing prompts.

2. The work demonstrates strong practical impact by showing that cache‑key collisions can reliably redirect multi‑step agent reasoning with high effectiveness, achieving up to 85% response hijacks and 80% tool‑use failures across realistic agentic settings. The inclusion of a concrete collision‑path example and evidence of cross‑model transferability further highlight the depth of the contribution, indicating that the study spans conceptual vulnerability analysis, rigorous threat modeling, and compelling applied validation.

3. The technical formulation is both thorough and well‑structured, clearly detailing the generator–validator architecture, the collision‑driven optimization objectives, the relaxed LSH gradient formulation for KV‑cache attacks, and the statistical validation mechanism used to classify cache hits under noisy latency signals. The distinction between CacheAttack‑1 and CacheAttack‑2 is especially clear, with the latter’s surrogate‑assisted filtering improving feasibility and reducing timing‑based detectability, indicating that the work reflects careful engineering judgment, explicit design trade‑offs, and a mature understanding of practical attack constraints.

4. The paper has no clear conclusion section and ends abruptly. A conclusion plays an important role in summarizing the findings, reinforcing the central contributions, and giving readers a coherent sense of closure. Without it, the manuscript feels unfinished, and key takeaways that should be consolidated at the end are left implicit rather than explicitly stated.

5. There is no limitations section, which makes it difficult to understand the boundary conditions of the approach or the scenarios where it may be less effective. A dedicated limitations discussion would help contextualize the system’s assumptions and clarify how broadly the proposed attack framework can be applied.

6. The paper does not include baseline comparisons, limiting the ability to contextualize the method’s performance relative to alternative approaches. Without comparisons to prior work or standard baselines, it is difficult to assess whether CacheAttack represents a significant empirical improvement over existing or simpler attack strategies.

7. While the related work material aligns well with the paper’s contributions, it is combined with the background rather than separated into its own section. This structural choice can disrupt the expected narrative flow, as it makes it harder to distinguish foundational background material from prior research directly relevant to positioning the paper’s contributions.

---

> ### Author Rebuttal · Authors · 2026-03-31
>
> Thanks a lot for your valuable review.
>
> - **Q1 & Q3: Adding Conclusion & Limitations**:
> In the final version, we will include a dedicated **Conclusion** section to synthesize our findings and a **Limitations** section. Specifically, the Conclusion will explicitly summarize:
>     - the newly identified attack surface (key collisions in semantic caching),
>     - the effectiveness across agentic pipelines, and
>     - the implications for real-world deployment.
> - **Q2: Comparisons with existing attack methods**:
> We would like to clarify that this work presents the first systematic study of key collision attacks on LLM semantic caching (as explicitly stated in the abstract, introduction, and contributions). As a result, there are no directly relevant baselines in the literature for the attack surface we study.
> The closest attack in the literature is PoisonedRAG[1], which we already discussed and distinguished in Sections 2.4 and 4.2. In **PoisonedRAG**, the attack targets the RAG system, including both the **retriever** and the **knowledge bas**e. This is fundamentally different from our **threat model** in terms of both the **attacker’s capability** and the **attack vector.**
>     - **PoisonedRAG (Data Poisoning):** This attack relies on **write access** to the external **knowledge base**. Although they define "black-box" as having no knowledge of the **retriever**’s internal parameters, the adversary must still **inject malicious entries** within the knowledge base ***before*** inference to subvert the retrieval results.
>     - **Our Attack (Inference-time cache key collision):** In stark contrast, our attack is **strictly inference-time and observation-only**. We assume the adversary has **zero write access** to the cache base or the system configuration. The attacker acts as a standard end-user, solely submitting prompts and monitoring **side-channel signals** (e.g., latency and response patterns) to exploit the semantic caching mechanism.
>
>     Furthermore, the two attacks differ fundamentally in their **operational timing** and **dynamism**. In **PoisonedRAG**, the RAG knowledge base functions as a **static** component during inference: malicious texts are injected before any user queries arrive, so the retrieved content remains unchanged regardless of subsequent inputs. In contrast, **semantic caching** is inherently **dynamic**—cache keys are generated and matched **on-the-fly** for every incoming query at inference time. This runtime nature allows **CacheAttack** to trigger false-positive key collisions purely through adversarial prompts, without any pre-injection into or modification of a persistent knowledge base.
>
> - **Q4: Structural Separation of Section 2**:
>     We will restructure Sections 2 and 3 to clearly distinguish our foundational concepts (fuzzy hashes) from prior research in privacy and side-channel attacks.

---

> > ### Author Rebuttal · Reviewer_8ihE · 2026-03-31
> >
> > With these revisions and clarifications, my earlier concerns have been resolved, and my overall assessment of the paper remains positive. Although the lack of directly comparable baselines is understandable, I encourage the authors to include simple reference‑point baseline comparisons in the final version to help contextualize CacheAttack’s empirical performance relative to more naive or approximate alternatives.

---

> > > ### Author Response · Authors · 2026-04-04
> > >
> > > Thank you for the constructive suggestion. Our attack method addresses a constrained black-box prompt optimization problem: to find improved prompts in a continuous embedding space while preserving semantic similarity to the original prompt, using surrogate modeling and gradient-guided optimization.
> > >
> > > Motivated by this design, we choose two complementary derivative-free baselines.
> > > First, synonym substitution represents the most direct semantics-preserving local editing strategy, allowing us to test whether simple conservative modifications are already sufficient under the similarity constraint.
> > > Second, a genetic algorithm[1] represents black-box search without surrogate modeling or gradient information, allowing us to test whether the observed gains come merely from stronger search rather than from surrogate-guided gradient optimization.
> > >
> > > We apply these baselines to RQ1 (LLM Response Hijacking). Synonym Substitution yields a hit rate of 0.0%, while the Genetic Algorithm (GA) achieves 36.2%. In contrast, CacheAttack-1 achieves 86.9%. These results demonstrate that these two simple baselines are insufficient to reliably trigger collisions in embedding space.
> > >
> > > References:
> > >
> > > - [1] [https://arxiv.org/pdf/1804.07998](https://arxiv.org/abs/1804.07998)

---

### Decision · Program_Chairs · 2026-04-30

**Decision:**

Accept (regular)

**Comment:**

This paper studies integrity risks in semantic caching for LLM systems and proposes CacheAttack, a framework for exploiting key-collision vulnerabilities under realistic black-box settings. The work is timely and addresses an important but underexplored aspect of LLM deployment security. Reviewers broadly agree that the conceptual framing of semantic cache keys as fuzzy hashes is insightful, and the proposed attack is well-designed with strong empirical validation across multiple settings, including agent workflows and real-world systems.

While some concerns were raised regarding threat model assumptions, evaluation completeness, and presentation, the authors provided detailed rebuttals, including additional experiments on commercial systems and clarifications on attack feasibility, robustness, and differentiation from prior work. These responses significantly strengthened confidence in both the practical relevance and technical soundness of the work.